# Health Literacy, Social Networks, and Health Outcomes among Mental Health Clubhouse Members in Hawai‘i

**DOI:** 10.3390/ijerph20010837

**Published:** 2023-01-02

**Authors:** Joy Agner, Monet Meyer, Tyra Mahealani Kaukau, Maileen Liu, Lisa Nakamura, Adriana Botero, Tetine Sentell

**Affiliations:** 1Chan Division of Occupational Science and Occupational Therapy, University of Southern California, Los Angeles, CA 90089, USA; 2Department of Psychology, University of Hawai‘i at Mānoa, Honolulu, HI 96822, USA; 3Hawai‘i School of Professional Psychology, Chaminade University of Honolulu, Honolulu, HI 96816, USA; 4Office of Public Health Studies, Thompson School of Social Work, University of Hawai‘i at Mānoa, Honolulu, HI 96822, USA

**Keywords:** health literacy, serious mental illness, clubhouse, mental health clubhouse, social networks, distributed health literacy, community mental health

## Abstract

Health literacy is the ability to obtain and utilize health information to make health-related decisions and to navigate health systems. Although health literacy has traditionally been understood as an individual-level construct, current research is revealing the impact that social networks can have on health literacy. To date, no studies have examined associations between health literacy and social networks among people with serious mental illness (PWSMI), who are at high risk of physical illness and premature mortality. To begin to fill this gap, this study explores associations between health literacy, relationships with health discussion partners, and self-reported health outcomes in a racially diverse sample of Clubhouse members in Hawai‘i. Clubhouses are community mental health centers that promote recovery from mental illness through destigmatization, meaningful activity, and strong social relationships. Health literacy was assessed using two single-item screeners (SILS). In a sample of 163 members, 56.2% reported adequate ability to understand health-related instructions or pamphlets, and 43.3% reported adequate confidence filling out medical forms independently. This is consistent with other health literacy studies with PWSMI in the United States, and indicates lower health literacy within this group than is reported in national averages. Multivariate logistic regression revealed a larger Clubhouse staff social network and completing high school were significantly associated with requiring less help to read materials. Higher age, male gender, and being Native Hawaiian and/or Pacific Islander were associated with less confidence filling out medical forms, while higher self-efficacy was associated with higher confidence filling out medical forms. This study provides preliminary evidence that relationships fostered within Clubhouses are associated with health literacy among PWSMI, and highlights the need for more research to examine how social networks and health literacy interventions can be leveraged in community mental health settings to improve health outcomes within this vulnerable population.

## 1. Introduction

### 1.1. Health Literacy and Health Disparities among People with Serious Mental Illness

Individuals with serious mental illness (SMI) frequently suffer from chronic health conditions, and have dramatically lower life expectancy than the general population [1]. SMI is defined by severe functional impairment and most commonly includes the diagnoses schizophrenia, bipolar disorder, and major depression. It affects around 14.2 million Americans, and prevalence is increasing. From 2016–2020, prevalence grew from 4% to 5.6% of the general population [2]. Adults with SMI are more than twice as likely to be diagnosed with cardiovascular disease than those without [3,4], and 75% have at least one chronic illness in addition to their psychiatric diagnosis [5]. Effective and consistent engagement in health care, including access to relevant, trusted health information for both treatment and management of health conditions, is crucial to address health disparities faced by PWSMI [6], both in terms of preventing and managing chronic disease. As such, it is essential to examine health literacy among PWSMI.

Health literacy is the ability to obtain, understand and use information to engage in healthcare and prevent or manage illness [7]. It comprises the ability to read or understand written materials, numeracy skills, as well as basic familiarity with medical language and health care systems. Past research has shown that low health literacy is associated with poorer health outcomes as well as differences in health service utilization. In a systematic review, patients with poorer health literacy were at increased risk for hospitalization and emergency service while being less likely to utilize preventative health services [8]. This review also found less medication adherence and higher mortality among those who had poorer health literacy.

Research on health literacy among PWSMI, and its associations with their health or health service utilization, is very limited despite the vulnerability of this population. The sparse research that exists on health literacy among PWSMI in the United States (US) has found PWSMI have lower health literacy than the average population [9,10]. Analysis of the National Adults Literacy Survey showed individuals who reported a mental health problem were significantly more likely to report low health literacy after controlling for education and race [11]. Nationally, there are around 80 million individuals who have limited or low health literacy [12]. According to Kutner and colleagues [12], around 36% of the US population reports inadequate health literacy, while multiple studies focused on PWSMI have found higher rates of inadequate health literacy on average, using diverse measures. For example, Clausen and collaborators [9] examined the prevalence of low literacy among 71 PWSMI in a psychiatric rehabilitation program in Nebraska. Using a single-item literacy screener (SILS), they found that 42.3% had inadequate health literacy. Using a functional measure, the Rapid Estimate of Adult Literacy in Medicine (REALM-SF), 51.7% had 8th grade or below literacy. Within this sample, lower health literacy was associated with lower general functioning. A study by Krishan et al. [10] also found that PWSMI had below average rates of health literacy. In their sample of 256 adults with severe mental illness in Atlanta, 46% had low health literacy, which was associated with increased likelihood of inpatient medical hospitalizations. These studies highlight the need to further examine health literacy in this vulnerable group.

The following section reviews individual and social context factors associated with health literacy, followed by background on the setting for this study, mental health Clubhouses, and study aims.

### 1.2. Individual and Social Factors Associated with Health Literacy

*Individual factors*. A number of interrelated psychological and interpersonal factors are associated with health literacy. Some individual factors that have been shown to be associated with health literacy are gender, age, education, race, and psychological factors, such as stigma, shame and self-efficacy. In the United States, women tend to have higher health literacy than men [13,14]. Younger people also tend to report higher health literacy [15] as do individuals who have higher income or education levels [16,17,18].

Race has also been associated with health literacy. In a systematic review, African Americans were shown to have poorer health literacy than whites in the United States [19]. In a statewide study in California, Asian Americans and Pacific Islanders (AAPI) as well as Latinx participants had the lowest health literacy scores on average [20]. Studies within Hawai‘i highlight the importance of disaggregating AAPI, when possible, to identify subgroup differences. Sentell and colleagues [21] found in a statewide survey in Hawai‘i that Filipinos reported low health literacy most frequently (23.9%) compared to other racial/ethnic groups, followed by other Pacific Islanders (20.6%). Among studies considering health literacy with PWSMI, few studies have examined associations with race, and to our knowledge, none have included Native Hawaiians and other Pacific Islanders. Identifying these differences can be important for building trusted, culturally-relevant materials to support health equity, building from community strengths and perspectives.

In addition to racial disparities, and differences in age, income, and gender, past studies have found that shame, stigma, and self-efficacy are related to health literacy. Past research has found lower health literacy can result in shame when engaging in medical care, and can lead to individuals hiding their difficulty interpreting medical information or not seeking care altogether [22]. Stigma and shame are closely related concepts. Shame can be the internalized result of stigma, which tends to stem from societal prejudices [23]. PWSMI may experience multiple, intersecting types of stigma from the presence of mental illness and discrimination experienced as a result, the presence of any concomitant physical illnesses or poverty (which are common among PWSMI) and low health literacy. These compounding types and sources of stigma may inhibit individuals who could greatly benefit from engagement in health care from seeking the care they need. Conversely, self-efficacy, the belief that one is capable to make and execute decisions for themselves to succeed in different environments and situations [24], has been associated with higher health literacy [24,25]. Past research has shown that stigma and self-efficacy are inversely related [25,26,27], and that PWSMI tend to have lower self-efficacy than the general population, likely because of the stigma they regularly face [28]. To our knowledge, no studies have examined the relationship between self-efficacy and health literacy among PWSMI.

*Social environment, social networks, and distributed health literacy.* In addition to individual factors, social factors can have a profound effect on health literacy [29,30,31]. Increasingly, there is a recognition that health literacy, the ability to access and utilize health information, can be shared within a social network. Distributed health literacy is defined as the way health-related knowledge and information is shared among social networks [32]. In a longitudinal, qualitative study of health literacy in South Wales, researchers found that individuals relied on the health literacy skills of their family members and friends to engage in health-care decision making processes and to communicate with health professionals [32]. Similarly, Nimmon and Regher [33] studied health communication social networks longitudinally, and found that both the structure and the function of health communication networks changed over time. They found that physicians typically were one node of many in health communication networks, and that individuals frequently engaged individuals close to them in making meaning of illness experience, determining adherence to medical advice, and deciding on next steps. A social network perspective may be particularly important for addressing health inequities among cultural groups with a communal perspective on health decision making and information sharing, which include Native Hawaiian, other Pacific Islander, and Filipino populations [30,34,35].

Although no studies on distributed health literacy have been conducted with PWSMI, other studies have considered the role of social networks on health and healthcare engagement among PWSMI. In a study by [36], PWSMI who were connected to people who had trusting opinions of doctors had higher self-rated health and more regular engagement in health care. There have also been a variety of studies that suggest social networks and the presence of supportive relationships may aid in the recovery process of those with severe mental illnesses [37,38,39], and that smaller social networks are associated with more frequent psychiatric hospitalizations [40]. Based on the growing research on distributed health literacy, the dramatic health disparities experienced by PWSMI, closer attention should be paid to the ways in which supportive relationships among PWSMI might affect health literacy. One community mental health setting that fosters the development of close, lasting social relationships for PWSMI are mental health Clubhouses.

### 1.3. Clubhouses: Socially-Oriented Community Mental Health Centers

Clubhouses are unique, psychosocial rehabilitation settings that were created by and for PWSMI. They focus on the development of relationships and engagement in community through the work-ordered day [41,42,43]. The work of the Clubhouse is organized by units, such as a clerical unit or a kitchen unit, where the members and staff work side-by-side. PWSMI that attend Clubhouses are referred to as “members” and Clubhouses around the world adhere to a set of standards [44]. Members often have long-standing engagement in Clubhouses and many describe it as a family-like atmosphere [45,46]. Professional divisions between members and staff are intentionally minimized, there are no staff-only spaces within the Clubhouse, and members and staff often freely share personal information about health, work, and family [47,48,49]. Minimized boundaries between staff and members aid in reducing stigma and create the opportunity for reciprocally supportive relationships within Clubhouses [45].

Clubhouses are well researched for their impact on psychological health. They have been shown to reduce hospitalizations, reduce symptoms, and promote recovery [41]. However, no studies have examined the impact that relationships fostered by Clubhouses impact health literacy. Because of the long-term relationships that are fostered within Clubhouses, they are an ideal setting to examine how relationships may impact health literacy among PWSMI.

### 1.4. The Present Study

Considering the dramatic health disparities experienced by PWSMI, and the potential for health literacy to enhance health outcomes within this population, this study explores both individual and social factors that have been shown to affect health literacy. Specifically, we address the following research questions: (1) What is the prevalence of adequate and low health literacy among Clubhouse members, and does it differ by individual factors including age, education, gender, race/ethnicity, stigma, and self-efficacy? (2) Are the number of health discussion partner relationships with staff or members associated with health literacy among Clubhouse members? And (3) Is health literacy associated with self-rated physical or mental health among Clubhouse members?

## 2. Materials and Methods

### 2.1. Overview

This study used a cross-sectional survey method design. The survey was developed with input by Clubhouse members and staff and vetted prior to data collection as part of an ongoing community-based participatory research collaboration. This partnership is intended to address a wide range of factors associated with health and well-being among Clubhouse members.

### 2.2. Sample and Procedures

Participants were recruited from nine Clubhouses across four Hawaiian Islands (O’ahu, Hawai‘i, Maui and Kaua’i). Inclusion criteria were that all participants were Clubhouse members, 18 years or older, proficient in English, and diagnosed with serious mental illness. Data were collected as part of a larger study between October and December 2019 through 23 site visits to Clubhouses. The consent form was read and reviewed verbally with each participant to introduce the purpose of the study and to ensure comprehension. Participants were informed of potential risks and mitigation strategies. Potential risks included distress when discussing their experiences as well as breach of confidentiality. To mitigate risk of distress, participants were reminded that they were fully free to skip questions, that their services in Clubhouse were not tied to their participation in the study in any way, and they were also provided resources to contact in case of distress. The study took place within a mental health psychosocial rehabilitation center, and as such, resources were readily available if needed. To mitigate the risk of breach of confidentiality, participants were not asked for personally identifying information. Paper surveys were stored in a locked filing cabinet, and online data was password protected and only accessible to the research team. After consent, participants could complete the survey online using the survey platform *Qualtrics*. Options were also provided to complete the survey on paper, or via interview with a trained research assistant. In the case of surveys completed on paper or via interview, the data were entered into Qualtrics by the research assistant. Survey times ranged between 20–60 min.

### 2.3. Measures

The survey included demographic variables, self-efficacy, stigma, mental and physical health, stigma, social networks inside and outside of the Clubhouse and health literacy. Further detail on each of the measures is provided below.

#### 2.3.1. Demographics

Demographic variables included age, education, gender, and race/ethnicity. Since the sample is relatively small, several categorical demographic variables were collapsed into fewer groupings to avoid an overfit model. Participants had the option to select more than one category for race/ethnicity to indicate their racial/ethnic identity. Across the sample, participants identified with thirteen categories of race and ethnicity, including an “other” option. Among the sample of 163 participants, 41.7% identified as multiracial. The top five most common race/ethnicity identities were used for analyses. These were Native Hawaiian and Other Pacific Islanders (NHOPI), Filipino, Japanese, Chinese, and white, which mirrors the majority populations in Hawai‘i [50]. Participants who did not select any of these five identities, and those who selected “other not listed” or “prefer not to answer” were included in an “Other” group.

In regression models, we included non-multiracial whites as the reference group, and kept all multiracial individuals in the groups they self-identified with. So, someone might indicate they are part Chinese, part Filipino and would be reflected by both of those racial/ethnic variables in the model. This follows arguments against limiting participants who identify as multiracial to a single group [51,52,53]. The rationale for using non-multiracial whites as the reference group is twofold. First, from a perspective of racial injustice and colonization, scholars have argued that phenotype, or “socially assigned race” is an important component of racial treatment and discrimination, that may be an important determinant of health [54]. Thus, including an individual that is multiracial Japanese, African American and white within a white cohort may mask some factors associated with race/ethnicity. Second, past research has demonstrated differences in health literacy between white and AAPI individuals [20].

Participants indicated their gender identity on the survey from a list of options including male, female, non-binary, transgender, a gender not listed, and prefer not to answer. These were collapsed into male and not-male, and male was used as the reference group as it was the largest group. Initial education options were 8th grade or below, 9th–11th grade, high school graduate or GED, some college, college graduate, graduate school. These were collapsed into those who had graduated high school or received a GED and those who had not. For education, not completing high school was used as the reference group. Age was treated as a continuous variable.

#### 2.3.2. Self-Efficacy

The General Self-Efficacy Short Form from the Patient-Reported Outcomes Measurement Information System (PROMIS) was used to measure self-efficacy. The General Self-Efficacy Short Form contains eight-items such as “How confident are you that you can find ways to manage stress?” Each item is answered based on a 5-point Likert Scale. The form was scored using a system available through *HealthMeasures.net* (*HealthMeasures Scoring Service, n.d*.) (https://www.healthmeasures.net, accessed on 20 January 2021). HealthMeasures calculates a T-Score metric, a standardized score that can be used to compare populations across studies and to the reference group. The average for the T-Score in the normed population is always 50 and the standard deviation is 10. In this sample, the internal consistency was high (Chronbach’s α = 0.92). This form is available to the public at *HealthMeasures.net* (https://www.healthmeasures.net, accessed on 20 January 2021).

#### 2.3.3. Stigma

The Neuro-QOL (Quality of Life) Item Bank v1.0 Stigma Short Form from PROMIS was used to assess stigma. This measure is suitable for application to multiple populations [55]. Example items include, “Lately because of my illness I felt left out of things” [55]. The Stigma Short Form was scored using the T-Score metric. Items in the measure exhibited good internal consistency (Chronbach’s ⍺ = 0.91). This form is available to the public at *HealthMeasures.net* (https://www.healthmeasures.net, accessed on 20 January 2021).

#### 2.3.4. Health Literacy

Health literacy was assessed using two single-item literacy screeners (SILS). Both items were developed to briefly assess health literacy in clinical settings [56] and have been used widely since with a variety of populations [57,58,59]. Because the questions address qualitatively different aspects of health literacy, and these were of specific interest to understand distinctly in this sample, both were included and analyzed separately.

#### 2.3.5. Social Networks/Health Discussion Partners

To measure social networks, participants were asked the following questions: (1) “Are there Clubhouse staff that you could talk to about mental or physical health problems if they came up?” (2) “Are there Clubhouse members that you could talk to about mental or physical health problems if they came up?” (3) “Are there people outside the Clubhouse, such as family or friends, that you could talk to about mental or physical health problems if they came up?” If the response to either of those questions was “Yes” participants were prompted to list the first names and last initials of the individuals they would go to for support. We heretofore refer to these individuals as “health discussion partners.”

#### 2.3.6. Mental and Physical Health

Two items from the CDC’s “Healthy Days” health-related quality of life measure (HRQOL-4) were used to assess self-reported mental and physical health [60]. Mentally healthy days were assessed with the item: “Thinking about your mental health, which includes stress, depression, and problems with emotions, for how many days during the past 30 days was your mental health not good?” Physically healthy days were operationalized by the question: “Thinking about your physical health, which includes physical illness, injury or pain for how many days during the past 30 days was your mental health not good?” These measures have been validated and used in a wide variety of nationally representative studies in the US [61,62].

### 2.4. Data Analysis

Data were exported from Qualtrics, cleaned and examined for outliers and missing data. No outliers were removed. Cases with missing data were excluded listwise. Based on a missing values analysis, no patterns were found between missing data and demographics. Most commonly, social network data was missing or incomplete. This is in line with past research showing social network data is often subject to more missing data and respondent burden than survey questions that do not require recall [63,64].

For social networks, the number of members, staff, and people outside the Clubhouse (such as family or friends) that each member relied upon as a health discussion partner were summed resulting in three values: (1) staff network size, (2) member network size, and (3) outside network size. Health literacy variables were converted to a binary format (low health literacy and adequate health literacy). For the first SILS, “How often do you need to have someone help you when you read instructions, pamphlets, or other written material from your doctor or pharmacy?” response choices “Never” and “Rarely” were coded as adequate health literacy, while “Sometimes,” “Often,” or “Always” were coded as low health literacy. With the second SILS, “How confident are you in filling out medical forms by yourself?” The responses “Quite a bit” and “Extremely” were coded as adequate health literacy. The responses “Not at all,” “A little bit,” and “Somewhat” were coded as low health literacy.

Two multivariable logistic regression models were run in SPSS to examine associations between the two health literacy screeners and individual factors (age, education, gender, race/ethnicity, stigma and self-efficacy), relationships with health discussion partners (staff, members, and outside individuals), and health outcomes (mentally and physically healthy days). Each model used a different health literacy screener. Model one focused on the self-reported need for assistance when reading instructions or pamphlets, and model two used confidence filling out medical forms independently. After the descriptive and inferential analyses were complete, three social network visualizations were produced in R using the *igraph* package. The visualizations depict health literacy and health discussion relationships between members and staff.

## 3. Results

### 3.1. Participant Demographic Characteristics and Average Health Literacy

The initial sample for the larger study included 217 Clubhouse members; 54 were removed due to missing data, which left 163 in the analysis. Of these 163 members, 99 (60.7%) reported activity limitations because of an impairment or health problem. Additionally, members reported several types of SMI diagnoses; 63 (38.7%) reported multiple diagnoses, 54 (33.1%) schizophrenia, 15 (9.2%) schizoaffective disorder, 12 (7.4%) bipolar disorder, 11 (6.7%) depression, 3 (1.8%) other diagnoses, 3 (1.8%) missing diagnoses, and 2 (1.2%) reported no diagnosis. Table 1 presents descriptive statistics on age, race/ethnicity, and gender in the total sample, grouped by adequate and low literacy for each of the SILS. Fifty-six percent of members reported an adequate ability to understand health-related instructions or pamphlets. Forty-three percent of members reported adequate confidence in filling out medical forms independently. Table 1 highlights some of the differences in the two health literacy screeners associated with demographic variables. For example, males were more likely to report needing help reading instructions or pamphlets than not needing help (54% versus 46%), but more likely to report confidence filling out medical forms independently (55% versus 45%).

### 3.2. Social Network Characteristics and Average Health Literacy

Table 2 presents the average count and range of named member, staff, or outside health discussion partners in the Clubhouse. As a reminder, outside health discussion partners may include any individual with whom the member discussed health matters, such as friends, family, medical providers or caregivers. Within this sample, Clubhouse members were most likely to nominate staff as health discussion partners, followed by individuals outside the Clubhouse, and then fellow members.

### 3.3. Health Literacy Associations with Individual and Social Factors

Table 3 outlines the multivariate logistic regression results of both models. Mental and physical health during the past month, self-reported stigma related to living with SMI, the number of Clubhouse member health discussion partners, and the number of non-Clubhouse health discussion partners did not significantly impact the odds of having adequate health literacy as measured by either health literacy screener.

Confidence filling out medical forms was significantly associated with gender, race/ethnicity, and self-efficacy. Those identifying as not-male were more likely to report confidence filling out medical forms than males (OR = 2.336, *p* < 0.05), as were individuals who reported higher self-efficacy (OR = 1.066, *p* < 0.05); while those identifying as Native Hawaiian and/or Pacific Islander were less likely to report confidence filling out medical forms than non-multiracial whites (OR = 0.331, *p* < 0.05). Older Clubhouse members were also less likely to report confidence filling out forms than their younger counterparts (OR = 0.966, *p* < 0.05). Never or rarely needing help to read instructions or pamphlets was associated with higher education (completing high school or GED) (OR = 3.740, *p* < 0.05) and reporting more Clubhouse staff as health discussion partners (OR = 1.419, *p* < 0.05).

#### Health Literacy and Staff Health Discussion Partner Social Network Visualizations

The significant association between the number of staff identified as health discussion partners and health literacy is illustrated in Figure 1. These visualizations also demonstrate the heterogeneity among Clubhouses in average size, average health literacy, and the number of members with staff health discussion partners.

Visual inspection of the health discussion partner social networks between staff and members illustrates the relationship between adequate health literacy and connection to staff found in the multivariate regression. In the smaller Clubhouse (Clubhouse 1), all members named at least one staff person as a health discussion partner, and overall, health literacy was high in this group. In larger Clubhouses (Clubhouse 2 & Clubhouse 3) it is clear that some staff are more central in the network, and Clubhouse 3 appeared to be denser, with more members connected to particular staff. Additionally, Clubhouse 2 had seven members who reported no staff members they could talk to about health concerns, and overall shows fewer connections per staff member. Clubhouse 3 had ten members who reported no staff health discussion partners, but each staff member was connected to multiple members. This may suggest that certain staff in Clubhouse 3 were more commonly relied upon to discuss health-related concerns. Clubhouse 3 also had more members who did not name any staff people as health discussion partners. This suggests future studies should examine the impact of centrality, betweenness, density, and size of the overall Clubhouse on distributed health literacy.

## 4. Discussion

This exploratory study examined the prevalence of adequate and low health literacy based on two SILS among Clubhouse members with SMI, as well as associations between health literacy and individual factors (age, education, gender, race/ethnicity, stigma and self-efficacy), relationships with health discussion partners (staff, members, and outside individuals), and health outcomes (mentally and physically healthy days).

Overall, we found that self-reported health literacy was lower than national averages, which is around 36% according to Kutner and colleagues [12]. However, our findings were similar to other research with PWSMI that have been included in health literacy studies in the United States [9,10]. Interestingly, low health literacy has not been found in all studies among PWSMI. A study that included individuals with schizophrenia or depression in Australia found that participants’ health literacy was comparable with that of the general Australian population and adequate health literacy among PWSMI was reportedly much higher than typically reported in US studies [65,66,67]. Based on the Test of Functional Health Literacy in Adults (TOFHLA) 93% of their participants with depression and 97% with schizophrenia had adequate health literacy. This suggests that health literacy is not inherently connected to severe mental illness and can vary depending upon situational or environmental factors. As such, closer examination of health literacy averages among PWSMI, as well as individual and interpersonal factors that affect it, can help guide intervention planning and ultimately improve health literacy and health outcomes among this vulnerable population.

Within our sample we also found differential outcomes based on the health literacy screener. We included both screeners and analyzed them separately because, qualitatively, they seemed to address different aspects of health literacy, as discussed in more detail below. Reviewing past research using SILS, we found variation in which SILS was used, the number of SILS questions used (from 1–3), and outcomes related to the SILS item. The first SILS, “How often do you need to have someone help you when you read instructions, pamphlets, or other written material from your doctor or pharmacy?” has been associated with emergency department utilization, patient activation, and quality of life [65,66,67]. The second SILS, “How confident are you in filling out medical forms by yourself?” has been related to physical and psychological wellbeing as well as subjective well-being [68,69]. Because so few studies have included all three of the SILS developed by Chew et al. [56], and those that do often report results in aggregate overall [66,70,71,72], future research may examine the impact of individual SILS on distinct populations or consider analyzing them separately. These can impact health care utilization and health information access differentially in ways that may be particularly relevant for those with SMI.

### 4.1. Associations with Health Literacy Defined by Confidence Filling out Medical Forms

Within our sample, confidence filling out medical forms was less likely among males and older adults with SMI. Again, these findings mirror past research. Clouston et al. and Crowe et al. [13,14] found men were less likely to have adequate health literacy. However, again, this finding has not been replicated in all countries. In recent studies within Ghana and Iran, men had higher overall health literacy compared to women [73,74]. This suggests that the associations between gender and health literacy may be overlapping with access to education or other third variables not accounted for in our models.

Considering these findings in light of health intervention possibilities, it is important to point out that both men and women with SMI experience early mortality and high likelihood of concurrent chronic conditions [1,75,76,77,78], and that among PWSMI women tend to have higher premature mortality in comparison to males. In a study using United States Medicaid claims data, standardized mortality ratios over a period of 9 years indicated that white women with psychiatric illness were 5.6 times more likely to die than the general population, while white men with psychiatric illness were 3.4 times as likely to die than the general population. This suggests that both men and women with SMI are in high need of a full range of interventions to address health disparities, including those targeting health literacy.

Younger people in our sample also reported higher confidence filling out medical forms, which is similar to past research [15]. Self-efficacy was also associated with higher confidence. Because self-efficacy is operationalized by confidence in a variety of medical related settings, this finding may primarily indicate overlapping constructs, which may be taken in to account in future studies utilizing the SILS.

Finally, individuals in our sample who identified as Native Hawaiian and/or other Pacific Islander reported lower confidence filling out medical forms compared to non-multiracial whites. These subgroup differences also align with past research, and highlight that groups may have intersecting vulnerabilities to health disparities based on having mental illness and one’s racial/ethnic group. In an interview study with Native Hawaiians and Pacific Islanders (NHOPI) using the Newest Vital Sign (NVS) to assess health literacy, Lassetter et al. [79] found that 45.3% of their participants had low health literacy, and that low health literacy was associated with obesity in the sample. Sentell et al. [21] found that low health literacy was associated with diabetes and depression among Native Hawaiians. Past research on social networks and health literacy among Native Hawaiians and Pacific Islanders has found that individuals from these communities engage in health literacy practices with others in their social networks, including family members, friends, and co-workers [80]. Leveraging these social networks to engage in health promotion and intervention from a strengths-based perspective can support cultural preferences and help address health inequities [34,35].

### 4.2. Associations with Health Literacy Defined by Needing Assistance Reading Pamphlets or Other Written Materials

Our other measure of health literacy captured the need for assistance reading instructions, pamphlets or other written materials. Within our sample, this measure of health literacy was associated with having more staff members as health discussion partners. That is, members who were able to name more staff that they would turn to for help with a health problem, also reported needing less help with reading medical materials. Because this study is cross-sectional, this association can have various explanations. It may be that having close interactions or relationships with staff affects the members’ health literacy, which could be explained by some of the contagion studies of social networks. Or it may be the case that members with higher health literacy are more likely to seek out relationships with staff. A study on health seeking, health literacy, and arthritis management also found that individuals with higher health literacy tended to seek out health information from medical professionals, while those with lower literacy were more likely to consult informal networks or media sources for information [81]. Similarly, it is unclear whether the higher health literacy led to engagement with specialists, or the confidence to do so, or whether talking to specialists about their health led to higher health literacy. Future studies could shed light on these questions using longitudinal modeling, such as stochastic actor-oriented modeling to link the development of relationships with changes over time, and perhaps use health literacy interventions within a tight knit group, to examine how knowledge and perceived efficacy can flow through the network. In an examination of health information reach and flow in an Aboriginal community in New South Wales, a sociometric study indicated that key individuals within a community, e.g., those who were higher on closeness centrality and betweenness, were key disseminators of health information and knowledge to Aboriginal groups [82]. These individuals were also nested within Aboriginal-controlled health organizations. This illustrates how relationships fostered through trusted community-based organizations, like Clubhouses, can be key for enhancing health literacy and health knowledge among hard-to-reach groups. This also points to the importance of building organizational health literacy, which includes ensuring the health literacy of the clinical workforce, especially in supportive Clubhouse like environments in which residents are turning to staff for engagement and support for health decisions and health information processing [83,84]. Finally, education, defined by having graduated from high school or having a GED was also associated with higher health literacy. This also mirrors past research [16,17,18], and highlights a particular need for health literacy interventions among groups with lower formal education.

### 4.3. Limitations and Opportunities for Future Research

There are multiple limitations to this exploratory study that point to new directions in health literacy research and the influence of social context among PWSMI. First are the limitations of the measures. The two SILS we used did not address health literacy objectively, only through self-report. However, a study by Wallace et al. [70] confirmed that multiple questions from longer health literacy measures, such as the TOFHLA and REALM, were not more effective than the single-item screeners at predicting health literacy. Nevertheless, future research in this area could include more robust, objective, and comprehensive health literacy measures.

Second, the social network measures could be greatly expanded. Our study only includes the size of health discussion partner networks, examines variations by role (staff, member, person outside the Clubhouse), and used a name generator to elicit health discussion partners. Past research has shown that using name generators tends to elicit smaller networks [85]. Ideally, future studies will conduct sociometric and egocentric social studies that examine how health literacy is contained or potentially influential within a network, how and whether health literacy can flow through a network over time, and how different types of network measures (such as hierarchy, closeness, centrality, or homophily) affect distributed health literacy. Finally, future relationships should distinguish between types of relationships. Here, we examine the role of health discussion partners within Clubhouses. However, future research could examine the relative influence of in-person networks, family networks, friend networks, or social media networks [86,87]. Collaboration between experts in social network research, data science, public health, mental health, and sociology, could advance this line of research.

Third, our study has limitations in terms of examining the influence of race/ethnicity. In a highly multiracial sample there are many ways of aggregating and disaggregating the population, that have been the subject of conversation and debate among health researchers and many communities, policy makers, and other stakeholders [88,89]. Standard methods currently used are not necessarily fully reflective of the influence of race/ethnicity on health. Particularly for a small sample with this high of a proportion of multiracial participants, lumping all multiracial individuals together and/or collapsing race and ethnicity identities into mutually exclusive groupings has significant limitations. Primarily, this can lead to unintentional racial miscategorization [51]. Furthermore, arbitrarily creating mutually exclusive groups disregards the concurrent influence of historical factors that affect health and culture [90], and research indicating that even among individuals who indicate a “primary” ethnic identity, dominant aspects of multiracial identity can shift based on situational priming related to expectations, stereotypes, cultural norms [91] and perceived discrimination [52], a phenomenon described as multiracial malleability. Furthermore, research directly consulting multiracial respondents on their preferences indicates they often prefer to have the option to indicate each aspect of their background, rather than being forced to choose [52,53]. This is an area where future designs can become more nuanced and the impacts of various methods to study health impacts and health literacy in multiracial populations, and the intersection of with other identities associated with low health literacy, can and should be examined more closely and with larger sample sizes. Experts in population health among multiracial populations, and those with an intersectional lens, could be particularly useful in contributing to this dimension of health literacy research. Perhaps the emerging worldwide networks of health literacy researchers [92] could be used to transcend disciplinary or geographic lines and address these gaps in future research.

### 4.4. Potential Utility of This Work

We anticipate this work may provide near-term benefit in the following ways. First, this may increase awareness of the importance of addressing health literacy among this population. During the literature review, we were surprised by the paucity of literature within the United States on health literacy among PWSMI in particular, especially considering their increased likelihood for health vulnerabilities. As such, this work may increase attention to the health literacy needs within this population. Second, highlighting the role that relationships can both support psychosocial models of recovery from mental illness. Recovery is a broad concept that focuses less on symptom remediation, and more on living a full and healthy life. Accessing a support network, and having health discussion partners to rely on, may support recovery through distributed health literacy. Increasingly, there is attention on how community support settings, such as mental health Clubhouses, might intersect with, complement, and enhance the work of traditional medical models to improve health. Because our team works closely with mental health Clubhouses, we can share the results of this study with them directly, and perhaps highlight mental health literacy as a need among members, as well as a unique way that relationships within Clubhouses can be leveraged further to support member health. Finally, this work highlights the potential benefit of Clubhouses in a unique way. Several studies have examined the impact of Clubhouses on relationship development and mental health recovery, but none have framed those relationships as a mechanism to achieve better health through distributed health literacy and access to health discussion partners.

## 5. Conclusions

Despite the growing awareness and literature on how social networks can impact health literacy, this is the first exploratory study, to our knowledge, to examine the associations between health literacy and social networks among PWSMI. Despite limitations due to its cross-sectional and exploratory nature, these preliminary findings underscore the need for health literacy interventions aimed at reducing disparities among populations with SMI who face substantial, multi-faceted, intersecting vulnerabilities. Furthermore, community-based settings, such as mental health Clubhouses, may be leveraged to deliver health literacy interventions, and to better understand the impact of relationships on distributed health literacy.

## Figures and Tables

**Figure 1 ijerph-20-00837-f001:**
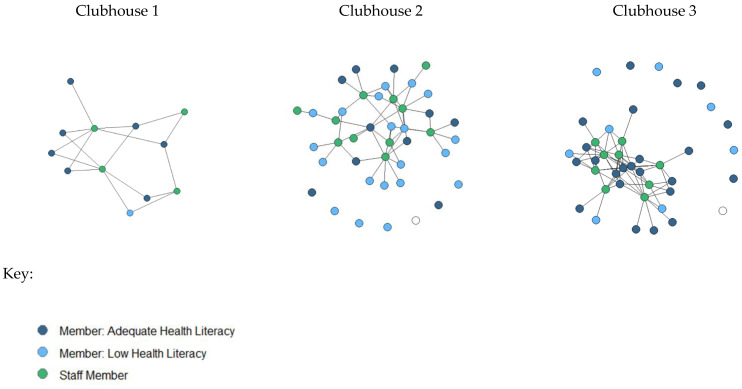
Visualization of Member-Staff Health Discussion Partner Social Networks and Health Literacy.

**Table 1 ijerph-20-00837-t001:** Participant Characteristics and Average Health Literacy.

		Needs Help Reading Instructions or Pamphlets		Confidence Filling Out Medical Forms Independently	
	Total (*n* = 163)	Low Literacy (*n* = 67)	Adequate Literacy (*n* = 96)	Low Literacy (*n* = 91)	Adequate Literacy (*n* = 72)
	Avg (SD)	Avg (SD)	Avg (SD)	Avg (SD)	Avg (SD)
Age	51.52 (12.52)	52.01 (13.85)	51.18 (11.56)	53.26 (12.26)	49.32 (12.58)
Gender	*n* (%)	*n* (%)	*n* (%)	*n* (%)	*n* (%)
Male	83 (50.92%)	34 (50.75%)	49 (51.04%)	52 (57.14%)	31 (43.06%)
Female	76 (46.63%)	31 (46.27%)	45 (46.88%)	36 (39.56%)	40 (55.56%)
Non-Binary	1 (0.61%)	1 (1.49%)	0 (0.00%)	1 (1.10%)	0 (0.00%)
Missing or Preferred not to Answer	3 (1.84%)	1 (1.49%)	2 (2.08%)	2 (2.20%)	1 (1.39%)
Race/Ethnicity *	*n* (%)	*n* (%)	*n* (%)	*n* (%)	*n* (%)
Native Hawaiian or Pacific Islander	54 (33.13%)	20 (29.85%)	34 (35.42%)	31 (34.07%)	23 (31.94%)
Filipino	43 (26.38%)	21 (31.34%)	22 (22.92%)	27 (29.67%)	16 (22.22%)
Japanese	34 (20.86%)	14 (20.90%)	20 (20.83%)	18 (19.78%)	16 (22.22%)
Chinese	33 (20.25%)	16 (23.88%)	17 (17.71%)	20 (21.98%)	13 (18.06%)
White **	22 (13.50%)	7 (10.45%)	15 (15.63%)	12 (13.19%)	10 (13.89%)
Other	24 (14.72%)	9 (13.43%)	15 (15.63%)	13 (14.29%)	11 (15.28%)
Missing or Preferred not to Answer	0 (0.00%)	0 (0.00%)	0 (0.00%)	0 (0.00%)	0 (0.00%)
Education	*n* (%)	*n* (%)	*n* (%)	*n* (%)	*n* (%)
Did not complete high school or GED	22 (13.50%)	14 (20.90%)	8 (8.33%)	13 (14.29%)	9 (12.50%)
Completed high school or obtained GED	141 (86.50%)	53 (79.10%)	88 (91.67%)	78 (85.71%)	63 (87.50%)
Missing or Preferred not to Answer	0 (0.00%)	0 (0.00%)	0 (0.00%)	0 (0.00%)	0 (0.00%)
	Avg (SD)	Avg (SD)	Avg (SD)	Avg (SD)	Avg (SD)
Stigma	54.51 (8.99)	56.38 (9.32)	53.20 (8.55)	54.68 (7.75)	54.28 (10.39)
Self-Efficacy	44.51 (12.37)	42.51 (12.73)	45.90 (11.99)	41.54 (12.06)	48.27 (11.81)
Number of Physically Unhealthy Days in Past Month	7.29 (9.84)	8.61 (10.53)	6.38 (9.27)	7.46 (9.84)	7.08 (9.91)
Number of Mentally Unhealthy Days in Past Month	8.06 (10.33)	11.06 (11.02)	5.96 (9.31)	8.82 (10.23)	7.08 (10.44)

* Participants could select more than one race/ethnicity. ** The only mutually exclusive grouping is the reference group, which includes non-multiracial whites.

**Table 2 ijerph-20-00837-t002:** Social Network Characteristics and Average Health Literacy.

		Needs Help Reading Instructions or Pamphlets		Confidence Filling Out Medical Forms Independently	
	Total (*n* = 163)	Low Literacy (*n* = 67)	Adequate Literacy (*n* = 96)	Low Literacy (*n* = 91)	Adequate Literacy (*n* = 72)
	Mean Count (Range)	Mean Count (Range)	Mean Count (Range)	Mean Count (Range)	Mean Count (Range)
Staff Network Mean Size (Range)	2.20 (0–9)	1.78 (0–8)	2.50 (0–9)	2.16 (0–9)	2.25 (0–9)
Member Network Mean Size (Range)	0.94 (0–12)	1.07 (0–12)	0.85 (0–5)	0.89 (0–12)	1.01 (0–5)
Outside Network Mean Size (Range)	1.47 (0–12)	1.45 (0–12)	1.49 (0–7)	1.48 (0–12)	1.46 (0–7)

**Table 3 ijerph-20-00837-t003:** Health Literacy Associations with Individual and Social Factors.

	Adequate Health Literacy			
Predictor Variables	Never or Rarely Needs Help Reading Instructions or Pamphlets		Quite a Bit or Extremely Confident Filling Out Medical Forms Independently	
	OR	95% CI	OR	95% CI
Gender **				
Non-male	1.53	0.71–3.27	2.34 *	1.10–4.97
Male				
Age	0.99	0.96–1.02	0.97 *	0.94–1.00
Race/Ethnicity				
Native Hawaiian or Pacific Islander	1.49	0.57–3.89	0.33 *	0.12–0.88
Japanese	1.13	0.46–2.76	1.17	0.48–2.80
Filipino	0.70	0.31–1.58	0.55	0.24–1.29
Chinese	0.62	0.25–1.59	0.88	0.35–2.23
Other	1.23	0.37–4.06	0.64	0.24–2.39
White (non-multiracial)				
Education				
High School Diploma/GED	3.74 *	1.21–11.52	0.81	0.28–2.36
Did not complete high school/GED				
Mental Health Not Good in Past 30 Days	0.97	0.93–1.01	0.99	1.00–1.03
Physical Health Not Good in Past 30 Days	0.99	0.95–1.03	1.00	0.96–1.04
Stigma	0.97	0.93–1.02	1.01	0.96–1.05
Self-Efficacy	1.01	0.97–1.04	1.07 *	1.03–1.10
Size of Clubhouse Member Social Network	0.78	0.61–1.02	1.00	0.78–1.28
Size of Clubhouse Staff Social Network	1.42 *	1.13–1.79	1.11	0.91–1.37
Size of Non-Clubhouse Social Network	0.95	0.76–1.18	0.90	0.72–1.14

OR = Odds Ratio; 95% CI = 95% Confidence Interval. ** p* < 0.05. ** As a reminder to the reader, non-male primarily includes female participants, and also includes non-binary, missing, and those who preferred not to answer.

## Data Availability

Data cannot be shared to protect privacy and confidentiality.

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
