# Peer review of "Health Literacy, Social Networks, and Health Outcomes among Mental Health Clubhouse Members in Hawai‘i"

_ijerph, 2023, doi:10.3390/ijerph20010837_

Round 1

Reviewer 1 Report

Health literacy is the ability to obtain and utilize health information to make health-related decisions and to navigate health systems. Although health literacy has traditionally been understood as an individual-level construct, current research is revealing the impact that social networks can have on health literacy. To date, no studies have examined associations between health literacy and social networks among people with serious mental illness (PWSMI), who are at high risk of physical illness and premature mortality.

The authors, to begin to fill this gap, this study explores associations between health literacy, relationships with health discussion partners, and self-reported health outcomes in a racially diverse sample of Clubhouse members in Hawai‘i. Clubhouses are community mental health centers that promote recovery from mental illness through destigmatization, meaningful activity, and strong social relationships.

The authors assessed health literacy using two single-item screeners (SILS). In a sample of 163 members, 56.2% reported adequate ability to understand health-related instructions or pamphlets, and 43.3% reported adequate confidence filling out medical forms independently, which reveals lower health literacy within this group than is reported in national averages. Multivariate logistic regression revealed a larger Clubhouse staff social network and completing high school were significantly associated with requiring less help to read materials. Higher age, male gender, and being Native Hawaiian and/or Pacific Islander were associated with less confidence filling out medical forms, while higher self-efficacy was associated with higher confidence filling out medical forms.

The authors concludes this study providing preliminary evidence that relationships fostered within Clubhouses are associated with health literacy among PWSMI, and highlights the need for more research to examine how social networks and health literacy interventions can be leveraged in community mental health settings to improve health outcomes within this vulnerable population.

Very interesting study.

I have only minor suggestions with a pure academic spirit.

1.     I like the purpose with questions “Considering the dramatic health disparities experienced by PWSMI, and the paucity of research with PWSMI study that addresses both individual and social factors that affect health literacy, this study explores multiple questions. 1) What is the prevalence of ade- quate and low health literacy among Clubhouse members, and does it differ by individuafactors including age, education, gender, race/ethnicity, stigma, and self-efficacy? 2) Are the number of health discussion partner relationships with staff or members associated  with health literacy among Clubhouse members? And 3) Is health literacy associated with self-rate physical or mental health among Clubhouse members?” I suggest to leave the question but introduce one or two sentences to explain the aim globally

2.     Avoid short paragraphs (se the par. 2.2)

3.     You say that the survey is online. Please describe in the methods the electronic design. Did you use Google forms, MS forms, survey monkey

4.     Please report in the results or in an appendix  the electronic survey

Author Response

Dear Reviewer,

Your insights have helped us make the work stronger and to provide clarification to future readers. Below, we address each of your comments in italicized text. You will also find modifications to the document in red track changes.

We sincerely thank you for taking the time to review this manuscript and provide feedback.

---------------------------------------------------------------------------------------------------------------------

Reviewer 1 comments and responses:

Very interesting study.

I have only minor suggestions with a pure academic spirit.

  1. I like the purpose with questions “Considering the dramatic health disparities experienced by PWSMI, and the paucity of research with PWSMI study that addresses both individual and social factors that affect health literacy, this study explores multiple questions. 1) What is the prevalence of ade- quate and low health literacy among Clubhouse members, and does it differ by individuafactors including age, education, gender, race/ethnicity, stigma, and self-efficacy? 2) Are the number of health discussion partner relationships with staff or members associated  with health literacy among Clubhouse members? And 3) Is health literacy associated with self-rate physical or mental health among Clubhouse members?” I suggest to leave the question but introduce one or two sentences to explain the aim globally

Thank you, a sentence was added to address the global aim, followed by the specific research questions.

  1. Avoid short paragraphs (se the par. 2.2)

This was corrected, and the rest of the manuscript was scanned for short paragraphs. Where possible, they were combined with others.

  1. You say that the survey is online. Please describe in the methods the electronic design. Did you use Google forms, MS forms, survey monkey

More details were added on this portion.

  1. Please report in the results or in an appendix the electronic survey

Because there were only a few measures from the larger study included in this analysis, we prefer not to upload the entire survey as an appendix. However, we have updated the manuscript so all items utilized in this study are available to a potential reader. The single items for demographics, health literacy, and social networks are all included in the text in the manuscript, and the two validated scales for stigma and self-efficacy are available free to the public through HealthMeasures.net. The web address to access these was added to the manuscript.

Reviewer 2 Report

This paper examines the associations between health literacy and social networks among people with severe mental illnesses (PWSMI). The present reviewer has no objection to the publication of this manuscript because the research methodology and the presentation of the reported scientific results have been adequately followed, for which the present reviewer requested consideration of the following issues:

1. Discuss the implications of this study in terms of the risks that may exist in the individuals who are the object of the study or in social terms, ranking them from highest to lowest impact.

2. Explain if tangible benefits are expected to be obtained in the short term from this research work? If so, describe what these benefits are in the short and medium term. Also describe possible alternative solutions to the proposal presented here.

3. Regarding the scope of this research, list other mental disorders that could be the subject of a similar study and justify your proposal.

4. Discuss if there are other disciplines that can help enhance this work in terms of conducting transdisciplinary research, eg data science.

5. Although this study is focused on the associations between health literacy and social networks among people with serious mental illnesses, it is suggested to the authors that other studies be included, such as those listed here: 

---The Role of Health Literacy and Social Networks in Arthritis Patients’ Health Information-Seeking Behavior: A Qualitative Study. Janette Ellis et al. International Journal of Family Medicine Volume 2012.

---Promoting Health Literacy in India Through Social Networks: Opportunities and Challenges. Narang Sangeeta et al. 2015, Volume : 53, Issue : 4

---Social media: A path to health literacy. Roberts, Michelle et al. Information Services & Use, vol. 37, no. 2.

--- Health literacy and social change: exploring networks and interests groups shaping the rising global health literacy movement.  Kristine Sørensen et al. Volume 25, Issue 4

Author Response

Dear Reviewer,

Your insights have helped us make the work stronger and to provide clarification to future readers. Below, we address each of your comments in italicized text. You will also find modifications to the document in red track changes.

We sincerely thank you for taking the time to review this manuscript and provide feedback.

---------------------------------------------------------------------------------------------------------------------

Reviewer 2

This paper examines the associations between health literacy and social networks among people with severe mental illnesses (PWSMI). The present reviewer has no objection to the publication of this manuscript because the research methodology and the presentation of the reported scientific results have been adequately followed, for which the present reviewer requested consideration of the following issues:

  1. Discuss the implications of this study in terms of the risks that may exist in the individuals who are the object of the study or in social terms, ranking them from highest to lowest impact.

A discussion of potential risks to participants in the study was included in the text, as well as steps that were taken to mitigate these. This was added to procedures near the section on informed consent.

  1. Explain if tangible benefits are expected to be obtained in the short term from this research work? If so, describe what these benefits are in the short and medium term. Also describe possible alternative solutions to the proposal presented here.

A section was added to the discussion on the potential utility of this work in the near-term. Thank you for this suggestion, as it is important to consider and articulate.

  1. Regarding the scope of this research, list other mental disorders that could be the subject of a similar study and justify your proposal.

We did not exclude any particular psychiatric diagnosis, and there was a wide variety within our sample. Although participants primarily had psychotic disorders, such as schizophrenia, there were also participants with mood disorders, such as depression and anxiety. Also, many participants had co-occurring psychiatric diagnoses. We added more information on the psychiatric diagnoses reported by participants in the results section.

  1. Discuss if there are other disciplines that can help enhance this work in terms of conducting transdisciplinary research, eg data science.

Areas where additional expertise could expand this work were added to the sections on “Limitations and Opportunities for Future Research.”

  1. Although this study is focused on the associations between health literacy and social networks among people with serious mental illnesses, it is suggested to the authors that other studies be included, such as those listed here: 

---The Role of Health Literacy and Social Networks in Arthritis Patients’ Health Information-Seeking Behavior: A Qualitative Study. Janette Ellis et al. International Journal of Family Medicine Volume 2012.

---Promoting Health Literacy in India Through Social Networks: Opportunities and Challenges. Narang Sangeeta et al. 2015, Volume : 53, Issue : 4

---Social media: A path to health literacy. Roberts, Michelle et al. Information Services & Use, vol. 37, no. 2.

--- Health literacy and social change: exploring networks and interests groups shaping the rising global health literacy movement.  Kristine Sørensen et al. Volume 25, Issue 4

Thank you for the suggestions. These citations have been added to relevant sections.